# Population exposure to multiple air pollutants and its compound episodes in Europe

Zhao-Yue Chen [1,2] ✉, Hervé Petetin[3], Raúl Fernando Méndez Turrubiates [1], Hicham Achebak [1,4], Carlos Pérez García-Pando [3,5] & Joan Ballester [1]

Air pollution remains as a substantial health problem, particularly regarding the combined health risks arising from simultaneous exposure to multiple air pollutants. However, understanding these combined exposure events over long periods has been hindered by sparse and temporally inconsistent monitoring data. Here we analyze daily ambient $PM_{2.5}$, $PM_{10}$, $NO_2$ and $O_3$ concentrations at a 0.1-degree resolution during 2003–2019 across 1426 contiguous regions in 35 European countries, representing 543 million people. We find that PM10 levels decline by 2.72% annually, followed by $NO_2$ (2.45%) and $PM_{2.5}$ (1.72%). In contrast, $O_3$ increase by 0.58% in southern Europe, leading to a surge in unclean air days. Despite air quality advances, 86.3% of Europeans experience at least one compound event day per year, especially for $PM_{2.5}$-$NO_2$ and $PM_{2.5}$-$O_3$. We highlight the improvements in air quality control but emphasize the need for targeted measures addressing specific pollutants and their compound events, particularly amidst rising temperatures.

Air pollution poses a major health risk in Europe and worldwide[1,2]. In 2021, the European Environment Agency (EEA) estimated over 253,000 premature deaths attributed to fine particulate matter ($PM_{2.5}$), 52,000 deaths to nitrogen dioxide ($NO_2$) and 22,000 deaths to ozone ($O_3$) exceeding the 2021 World Health Organization (WHO) annual limits[3]. These exposures, both chronic and acute, also increase the risk of cardiovascular and respiratory diseases, allergic reactions, diabetes, cognitive health, and childhood development, among many others[4,5]. Recognizing these risks, in 2021, the WHO[6] issued stricter air quality limits for each of these pollutants separately at different time scales, i.e. annual, peak season, 24 h and daily maximum 8 h, to mitigate both short-term and long-term health impacts caused by air pollutants.

To assess the threat posed by air pollution in Europe, recent compliance studies have predominantly relied on ground-based air pollutant monitoring networks[7–9]. However, these networks, concentrated primarily in urban areas, exhibit limited spatial coverage and fail to comprehensively represent the entire population. While ground-level measurements offer direct, accurate, and reliable real-world data,

their spatial averaging and extrapolation introduces biases in exposure assessment. Additional limitations include frequently incomplete daily observation time series values, which can lead to biases when averaging observations from varying numbers of sites per day. Also, data availability from these networks is higher in more recent periods[10], leading to inconsistencies in the prior analysis of multi-decadal concentrations changes.

Another key limitation pertains to the conventional analysis of guideline exceedances for each pollutant separately[7–9]. This approach overlooks occurrence of compound air pollution episodes, in which the WHO daily guidelines are simultaneously exceeded for two or more air pollutants. This is a noteworthy omission, as individuals may experience concentrations exceeding safe guidelines for multiple pollutants concurrently, potentially resulting in synergistic health effects that amplify overall health risks[11,12]. Although some have begun exploring the interactive health impacts of co-exposure to specific combinations of pollutants, such as $PM_{2.5}$ and $O_3$, further research on other combinations is imperative. Unfortunately, the unavailability of consistent daily ground-level measurements for multiple air pollutants

[1]ISGlobal, Barcelona, Spain. [2]Universitat Pompeu Fabra (UPF), Barcelona, Spain. [3]Barcelona Supercomputing Center, Barcelona, Spain. [4]Inserm, France Cohortes, Paris, France. [5]ICREA, Catalan Institution for Research and Advanced Studies, Barcelona, Spain. ✉e-mail: zhaoyue.chen@isglobal.org

presents a challenge in comprehending the spatio-temporal patterns of population's co-exposure.

Using models constrained with observations represents a promising solution to these problems[11]. Global atmospheric composition reanalyses provide multidecadal daily estimates integrating a diverse range of satellite measurements[12,13]. However, due to their coarse spatial resolution and the lack of integration of surface measurements, these datasets remain affected by significant biases at ground level. The use of air pollution models constrained by surface measurements over multidecadal periods, either for Europe or globally, have mostly focused on long-term averages (annual or monthly values)[14–16], while models predicting daily concentrations have predominantly focused on a single pollutant, primarily $PM_{2.5}$[17,18]. Consequently, consistent and accurate air pollution datasets allowing comprehensive understanding of population exposure to multiple air pollutants and its compound episodes in Europe is still lacking.

Governments worldwide are increasingly acknowledging the necessity of addressing air pollutions collectively, such as the integrated control programs in the United States[19,20], due to their cost-benefit efficiency, as well as the significant benefits they offer in improving overall air quality and public health. Unfortunately, the lack of spatial-resolved daily estimates over long period for multiple air pollutants obtained from internally-consistent models impedes our understanding of how multiple pollutants and their compound episodes have evolved over time in response to air pollution policies and measures implemented in Europe. Obtaining such crucial information is vital for evaluating the effectiveness of interventions and developing targeted strategies to mitigate the health risks associated with the exposure to multiple air pollutants.

This study uses Quantile LightGBM (QLG) machine learning models[21] to link ground-level station data of daily mean $PM_{2.5}$, $PM_{10}$, $NO_2$, and $O_3$ (the primary four air pollutants contributing to mortality[3]) concentrations with meteorological and air quality reanalysis data, aerosol optical depth (AOD) model estimations and ground-level emission data. The models estimate daily concentrations from 2003 to 2019 at a spatial resolution of 0.1°, which are then used to estimate regional population-weighted (PW) averages for 1426 NUTS3 regions in 35 European countries. These estimations were used to estimate the spatial heterogeneity and temporal evolution of (i) air pollution concentrations and the (ii) population count and (iii) cumulative time of exposure to concentrations exceeding the 2021 short-term and long-term WHO guidelines. Moreover, we analyzed the joint exceedance of WHO limits simultaneously for two or more air pollutants, providing a comprehensive assessment of compound events. This study contributes to the assessment of overall air quality in Europe within the framework of the new WHO short and long-term guidelines, and identifies spatiotemporal patterns of compound event days. This information is crucial for environmental health assessments and policymaking aimed at mitigating the health risks associated with air pollution in the European Union and the whole continent.

## Results
### Data validation
Our models demonstrate robust spatial cross-validation performance (see Fig. 1, Figs. S2, S3 in Supplementary Information) in estimating the European ground-level concentrations of $PM_{2.5}$, $PM_{10}$, $NO_2$ and the maximum daily 8h average of $O_3$ (here referred to as MDA8 $O_3$ for simplicity), with a NRMSE (Normalized Root Mean Square Error) of 1.85%, 2.71%, 8.99% and 3.20%, and a Pearson correlation of 0.80, 0.79, 0.79 and 0.90, respectively. $PM_{2.5}$ and $O_3$ estimations were nearly unbiased (NMB (Normalized Mean Bias) = −0.9% and 0.24%, respectively), while $PM_{10}$ and $NO_2$ were slightly underestimated (NMB = −3.81% and −2.00%, respectively). Table S4 (Supplementary Information) shows that the model estimations clearly outperform reanalysis data from CAMSRA[12] and MERRA-2[13], and Table S5 (Supplementary

Information) that the temporal cross-validation of the model estimates is consistent over the whole period. The comparison between whole-period averages of observed and estimated daily values is shown in Fig. S1 (Supplementary Information), and the mean, standard deviation, median, inter-quartile range, trend, and Pearson correlation in Northern, Southern, Western and Eastern Europe in Table S6 (Supplementary Information).

### Population-weighted concentrations and trends
Figure 2 depicts the long-term averages (left panels) and trends (right) in PW concentrations. Whole-period continental averages for $PM_{2.5}$, $PM_{10}$, $NO_2$ and MDA8 $O_3$ were 14.34, 22.01, 13.46, and 74.51 µg/m³, respectively. $PM_{2.5}$ and $PM_{10}$ was higher in Northern Italy and Eastern Europe, with high $PM_{10}$ additionally in Southern Europe. High $NO_2$ was mainly observed in Northern Italy and some areas of Western Europe, such as in the south of the United Kingdom, Belgium and the Netherlands. MDA8 $O_3$ was latitudinally orientated, with the highest concentrations in the Mediterranean. $PM_{2.5}$, $PM_{10}$ and $NO_2$ generally decreased in most of Europe, with an average annual rate of −1.72%, −2.72% and −2.45%, respectively. The most important reductions in $PM_{2.5}$ and $PM_{10}$ were observed in Central Europe, while for $NO_2$ they were found in mostly urban areas of Western Europe, which correspond to the areas with the highest concentrations. In contrast, MDA8 $O_3$ increased by 0.58% in Southern Europe, while it decreased or showed nonsignificant trend in the rest of the continent.

### Cumulative time of exposure
Figure 3 depicts the year-to-year time-series and whole-period average maps of the short-term unclean air exposure time (orange bars and maps) and the percentage of population living in short-term clear air areas (blue curves). Here, the annual unclean air exposure time represents the PW average annual number of days in which the WHO daily limit for an air pollutant is exceeded, while population in clean air areas represents percentage of people living in areas where air quality meets recommended standards. Detailed definitions are provided in the methodology section. Overall, we observed a consistent decrease in unclean air exposure time for $PM_{2.5}$, $PM_{10}$ and $NO_2$ throughout Europe, with approximately 60, 46, and 48 fewer unclean air days in 2019 compared to 2003, respectively. For MDA8 $O_3$, it generally increased, with the exception of the extreme year of 2003. In 2003, under the record-breaking summer temperatures[22], $O_3$ levels were similar to those registered within the period 2015–2019. Higher values of unclean air exposure time were found in Eastern Europe and Northern Italy for $PM_{2.5}$ and $PM_{10}$, in mostly urban regions (particularly in Western and Central Europe and Northern Italy) for $NO_2$, and in Southern and Eastern Europe for MDA8 $O_3$.

### Population in short-term or long-term exposure to clean air areas
As expected, in general terms, the trend of the annual percentage of population in short-term clean air areas was found to be negatively correlated with the evolution of the unclean air exposure time. Among the four pollutants, the population in short-term clean air areas for $PM_{10}$ exhibited the largest increase, rising from 8% in 2003 to 77% in 2019, an increase that represents 367.9 million people. The population living in clean air areas for $NO_2$ and $PM_{2.5}$ also increased from 21% to 49% and from 3 to 11%, respectively, corresponding to an increase of 141.2 and 41.8 million people compared to 2003, respectively. Changes in MDA8 $O_3$ exhibited significant annual variation. By 2019, the percentage of population in short-term clean air areas dropped from 62% to 26%, equivalent to around 219 million fewer people compared to 2004 (note: we here exclude 2003, which was an exceptional year in terms of O3 concentrations). Most short-term clean air areas were mainly located in Northern Europe, Scotland, the island of Ireland and Northern Spain.

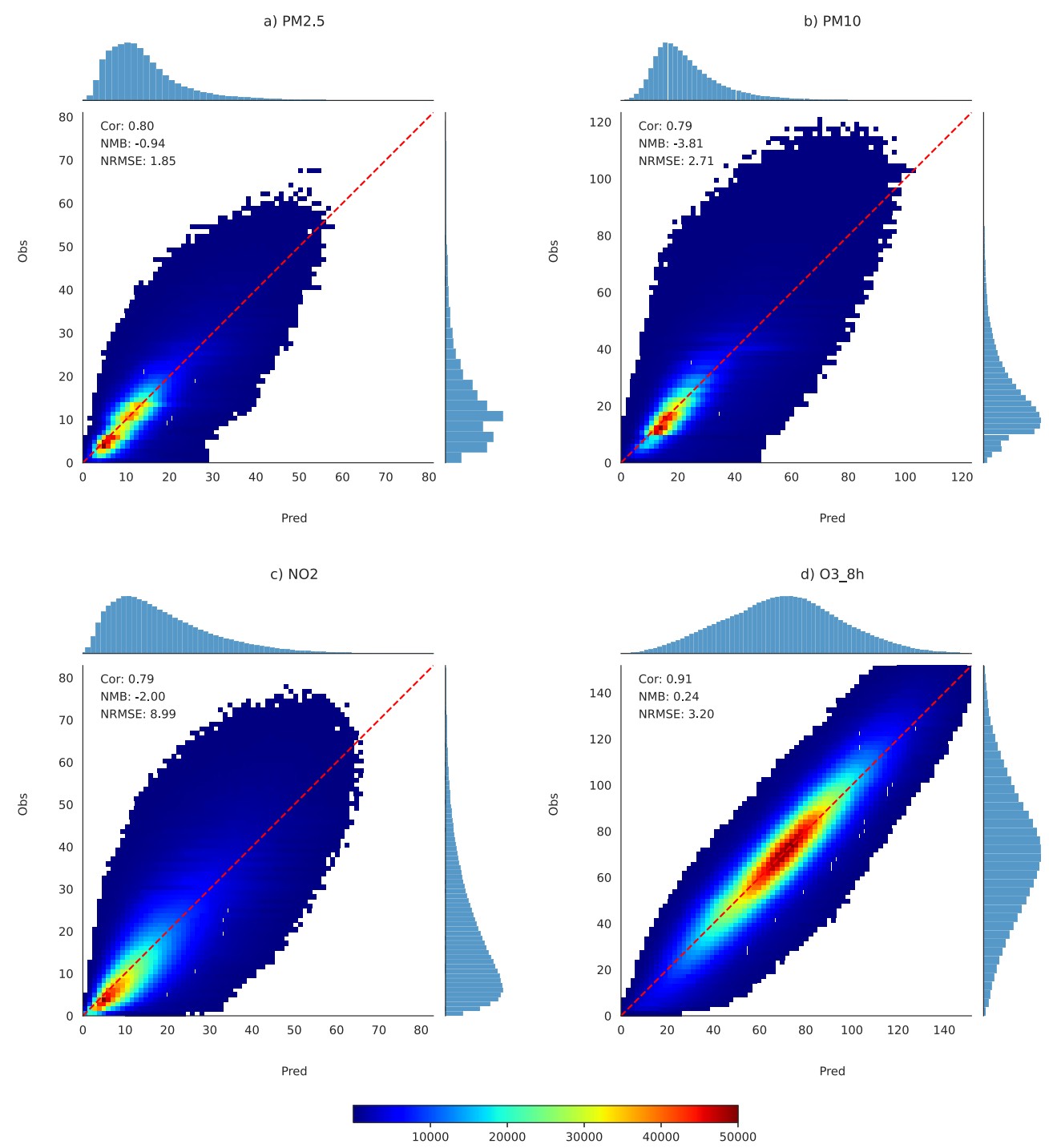

**Fig. 1 | Validation of estimated pollutants.** Comparison between observed and model-estimated PM$_{2.5}$ (**a**), PM$_{10}$ (**b**), NO$_2$ (**c**), MDA8 (maximum daily 8 h average) O$_3$ (**d**) concentrations from 2003 to 2019 under spatial cross-validation.

As a consequence of the general decline in air pollution levels, we found an increasing trend in the population residing in long-term clean air areas of PM$_{2.5}$, PM$_{10}$, and NO$_2$ (Fig. 4a–c), reaching 10.1, 102.8 and 75.9 million people in 2019, respectively. These numbers constitute about 1.90%, 19.85%, and 14.66% of the total population, compared to around 0.05%, 2.26%, and 2.88% corresponding to 2003. The distribution of the population living in long-term clean air areas is spatially heterogeneous. For example, the long-term clean air population for PM$_{10}$ and NO$_2$ grew faster in western Europe compared to other areas, while the increasing trend for PM$_{2.5}$'s clean air population was faster in northern Europe. The year-to-year changes in population

living in long-term clean air areas was found to be sometimes abrupt, as soon as densely populated areas started to comply with the regulation limits. Regarding MDA8 O$_3$, almost no areas meet the WHO standard of 60 µg/m3. Therefore, we adopted the WHO interim target 2 (70 µg/m3) as the reference threshold, which showed no clear trend over the years (Fig. 4d). Most long-term clean air population for MDA8 O$_3$ is found in western Europe.

**Population exposure to compound episodes**

In Fig. 5, we analyzed the annual unclean air exposure time for the joint exceedance of multiple air pollutants, or compound event days.

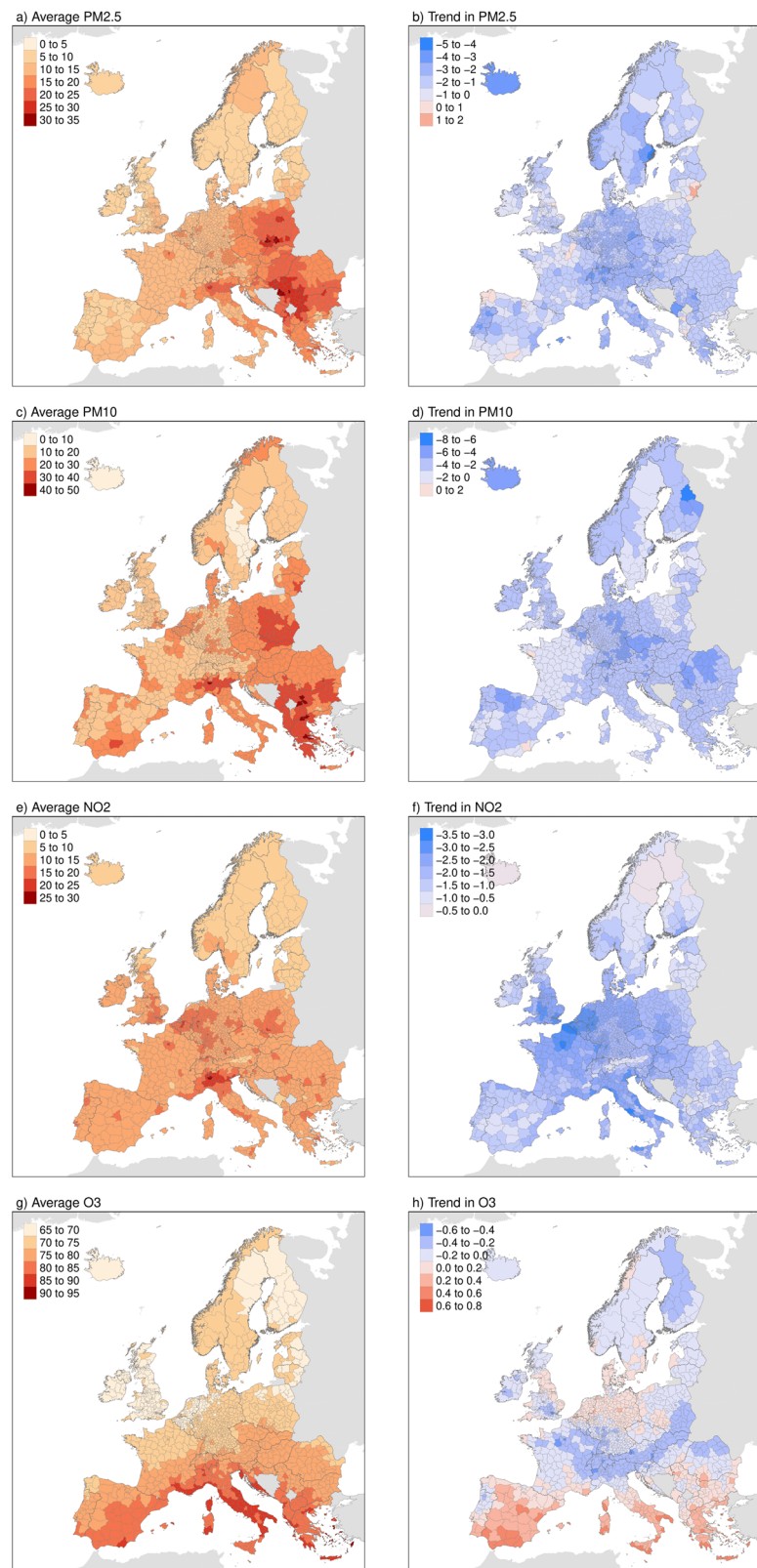

**Fig. 2 | Spatial Distribution and Trends of air pollution in Europe.** 17-year Mean Population-Weighted Concentrations of estimated PM$_{2.5}$ (**a**), PM$_{10}$ (**c**), NO$_2$ (**e**), and MDA8 (maximum daily 8h average) O$_3$ (**g**) (Unit: µg/m³), and its Average Annual percentages Changes (in %, calculated using Theil-Sen slope dividing mean estimates) (**b**, **d**, **f**, **h**).

Generally, the annual unclean air exposure time for various types of compound event days decreased significantly in Europe, dropping from 78.5 days to 17.4 days. During 2012–2019, around 86.26% of the European population experienced at least one day per year with compound event days, which is approximately 10% lower than the figures from 2003 to 2011 (see Table S7 in Supplementary Information). We identified four primary types of compound event days in Europe, namely PM$_{2.5}$-NO$_2$, PM$_{2.5}$-PM$_{10}$, PM$_{2.5}$-O$_3$ and PM$_{2.5}$-PM$_{10}$-NO$_2$

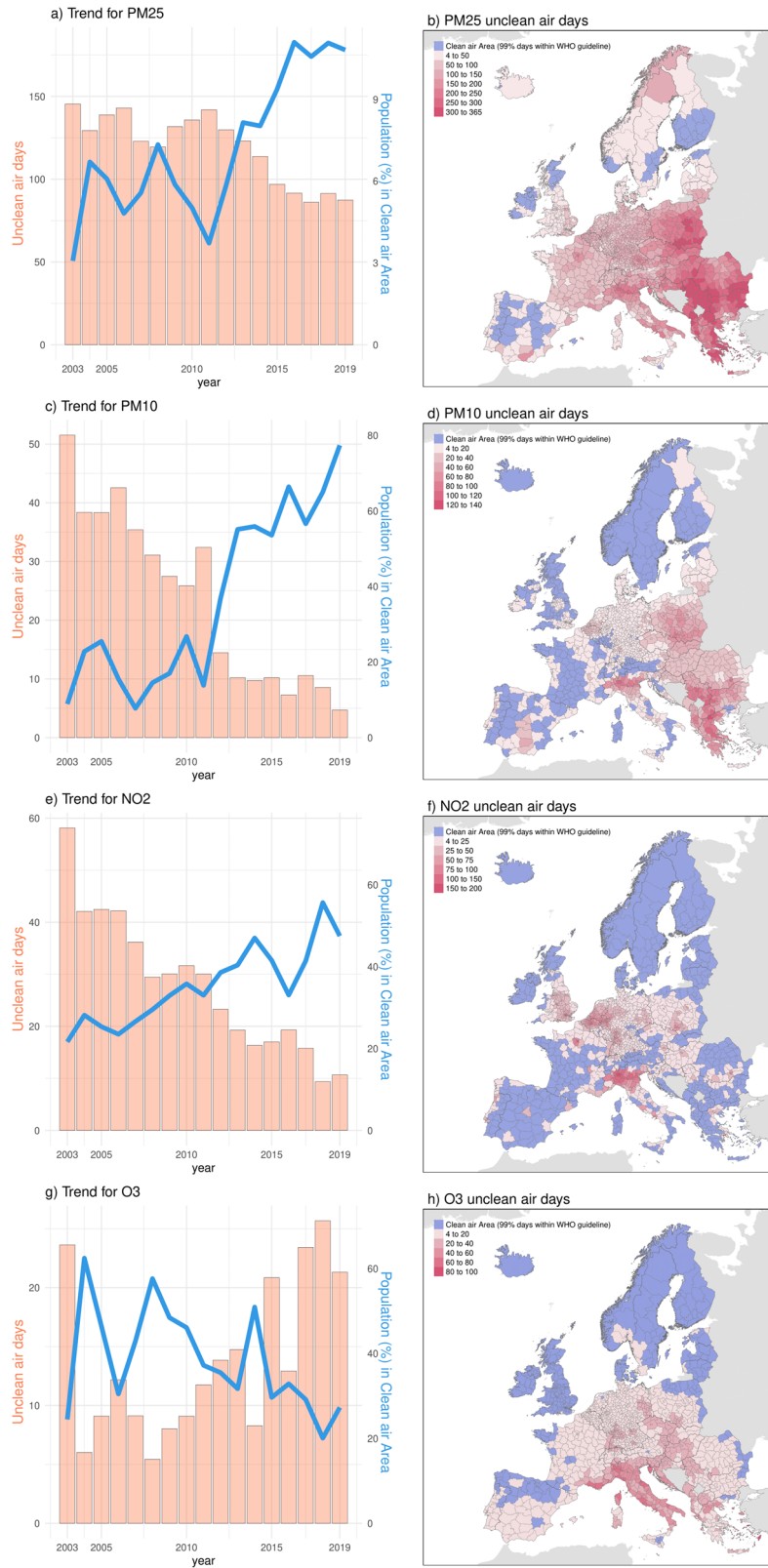

**Fig. 3 | Population exposure to short-term air pollutants in Europe.** Annual unclean air exposure time (Unit: Days, presented by bar plots) exceeding WHO Daily Limits, and the population (%) in short-term clean air areas (depicted by the blue line) for PM$_{2.5}$ (**a**), PM$_{10}$ (**c**), NO$_2$ (**e**), and MDA8 (maximum daily 8 h average) O$_3$ (**g**) in Europe. The spatial distribution of 17-years average annual unclean air exposure time (**b**, **d**, **f**, **h**). Short-term clean air areas here indicate those regions with 17-Year average annual unclean air exposure time Less Than 4 Days (WHO standard).

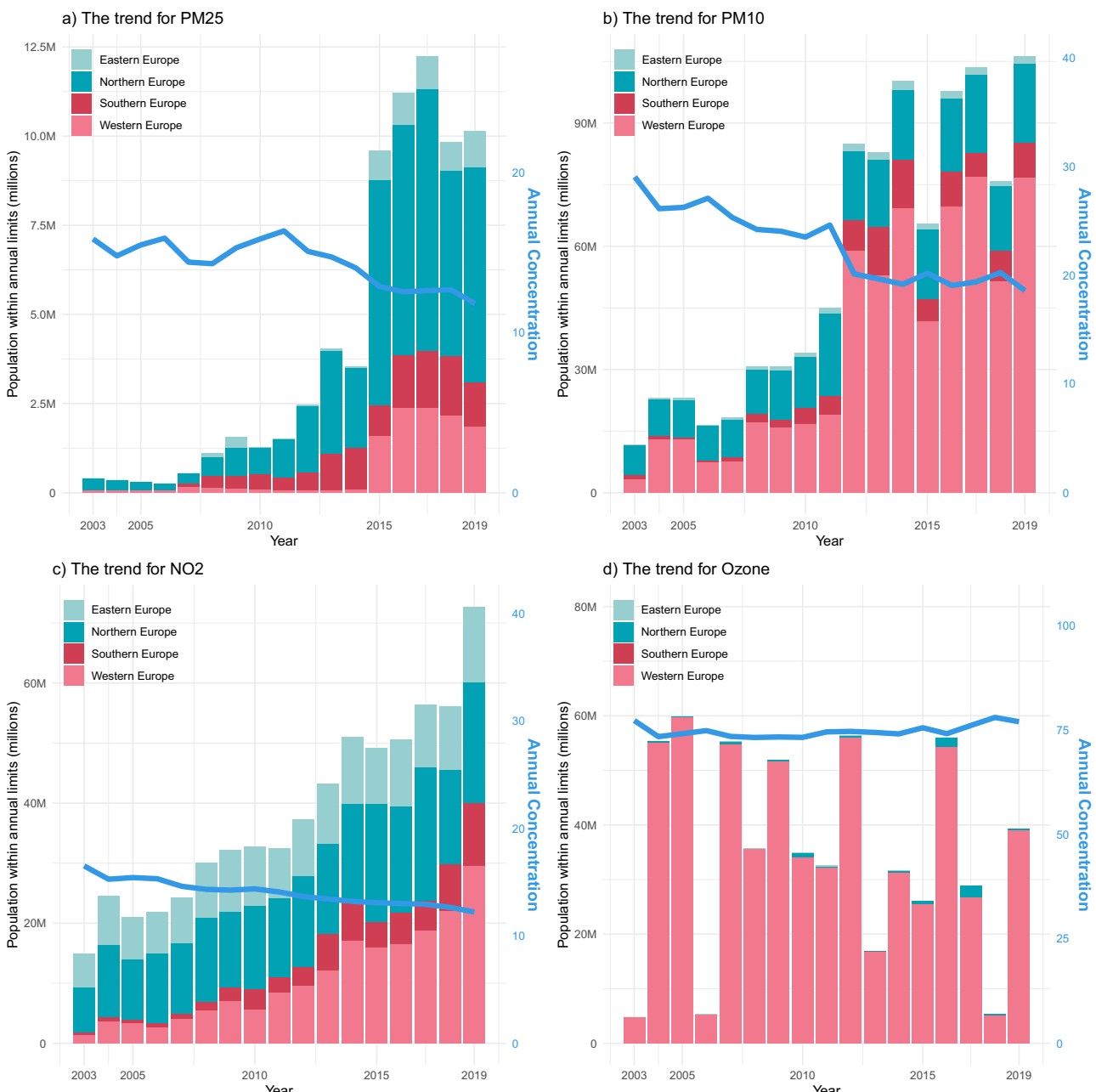

**Fig. 4 | Population exposure to long-term air pollutants in Europe.** Annual Population in long-term clean air areas (million, depicted by bar plots) and annual concentration (μg/m³, represented by blue line) in Europe for PM$_{2.5}$ (**a**), PM$_{10}$ (**b**), NO$_2$ (**c**), and MDA8 (maximum daily 8 h average) O$_3$ (**d**).

days, which collectively accounted for over 94.6% of all compound event days during the entire study period (see Fig. 5). Notably, compound event days also played a particularly important role in contributing to unclean air days, accounting for over 87% and 88% of unclean air days for PM$_{10}$ and NO$_2$, respectively (see Figure S4 in Supplementary Information).

The decreasing trends in compound event days are consistent across different subcontinental domain, although their contributions of compound event days vary (Fig. 5). Eastern Europe was found to be dominated by PM$_{2.5}$-PM$_{10}$ days, while PM$_{2.5}$-NO$_2$ days were more frequent in Western Europe. Southern Europe experiences a wider variety of types of compound event days, mainly PM$_{2.5}$-NO$_2$, PM$_{2.5}$-O$_3$ and PM$_{2.5}$-PM$_{10}$-NO$_2$. Although most compound event days are decreasing over decades, PM$_{2.5}$-O$_3$ days is the only one that increased (see Table s8 in Supplementary Information), going from 4.62 to 5.30 days per year when comparing the periods from 2012–2019 to 2003–2011. And

PM$_{2.5}$-NO$_2$ declined more slowly than other major types of compound event days. Consequently, PM$_{2.5}$-NO$_2$ and PM$_{2.5}$-O$_3$ days became the two predominant types of combinations in Europe during 2012–2019. Compound event days also exhibited a clear seasonal pattern in Fig. S5 (Supplementary Information). Compound event days with unclean levels of O$_3$ were more common from March to October, while those compounds involving PM$_{2.5}$, PM$_{10}$, or NO$_2$ tend to occur during colder seasons. Additionally, Figs. S6, S7 (Supplementary Information) illustrates a noticeable year-to-year decreasing trend in compound event days involving PM$_{2.5}$, PM$_{10}$, or NO$_2$.

## Discussion

In general, our study provides a comprehensive assessment of spatial and temporal inequities in population exposure to air pollutants in 1426 regions across 35 European countries, representing 543 million people. Our findings reveal a substantial reduction in European

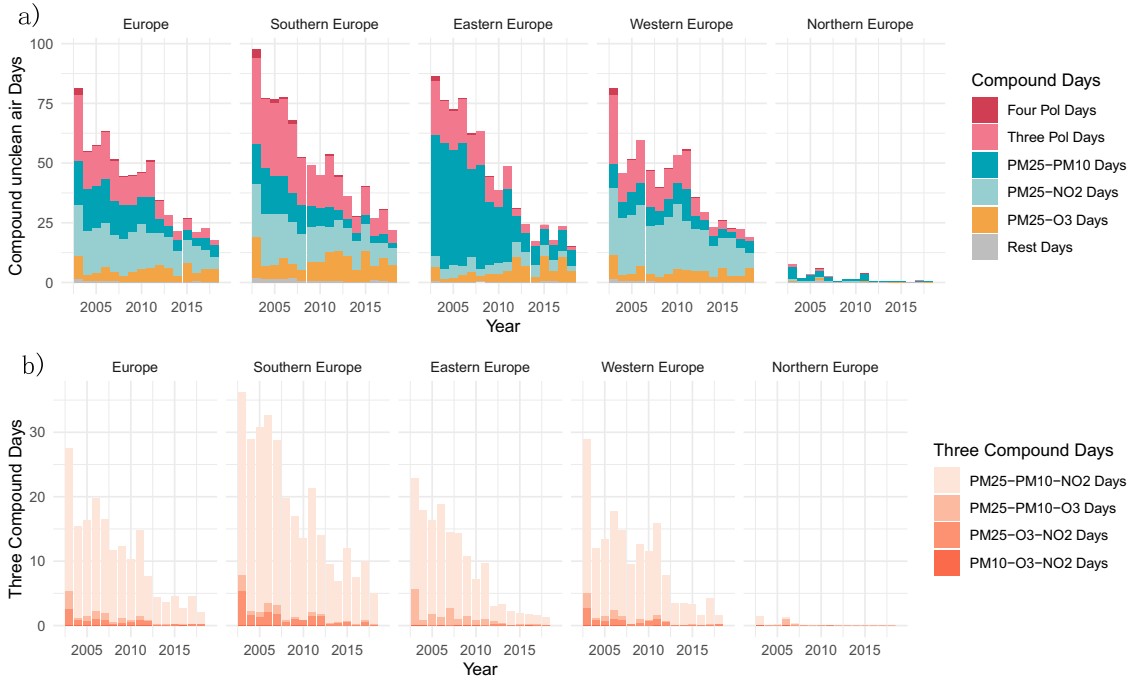

**Fig. 5 | Composition changes of multi-pollutant compound episodes in Europe.** The composition of annual compound unclean-air exposure days for multiple air pollutants across 17 years (**a**), with further focus on three pollutants combinations (**b**).

population exposure to most air pollutants. However, $PM_{2.5}$ and $O_3$ levels continue to surpass WHO guidelines in numerous regions, resulting in a relatively higher number of people exposed to unclean air levels. Moreover, our assessment of compound event days showed annual occurrence of compound event days decreased from 78.5–17.4 days over 2003–2019, but over 86.3% of the European population still experienced at least one compound event days per year in 2012–2019. $PM_{2.5}$-$O_3$ was the only compound event days that increased and became the second most frequent type of compounds in Europe during 2012–2019. Overall, our findings present comprehensive evidences of both short and long-term exposure to the main pollutants with largest impact on human health and mortality, by performing an exhaustive continental-wide regional analysis not restricted to urban settings only. Additionally, it introduces valuable insights into compound event days involving these pollutants, significantly enriching our understanding of multi-hazard exposure, and potentially guiding air pollution management policies.

Our research fills a critical gap in the literature by offering daily estimations of multiple air pollutants in Europe over the period 2003–2019, including $PM_{2.5}$, $PM_{10}$, $NO_2$, and MDA8 $O_3$. Unlike prior studies providing annual or monthly estimations over multidecadal periods[14–16], our daily air pollution estimations fill a crucial need for detailed data (either short-term or long-term) essential for conducting health impact studies and environmental monitoring. While prior studies focused mainly on single-pollutant estimations, mostly on $PM_{2.5}$[12,13], our study simultaneously provides estimations for multiple pollutants with enhanced predictive accuracy, achieving a strong correlation coefficient of approximately 0.79 to 0.90 for spatial cross-validation and 0.81 to 0.91 for temporal cross-validation. For example, Lary et al. estimated daily $PM_{2.5}$ concentrations globally over the period 1997–2014 by using remote sensing and meteorological data[17], with a correlation coefficient of 0.52–0.75. Moreover, Yu et al. used deep ensemble machine learning to estimate global daily $PM_{2.5}$ concentrations in 2001–2019[18], with Spearman correlation of around 0.76 with ground-level observations throughout Europe. Furthermore, our estimates also outperform CAMSRA and MERRA-2 reanalysis (see Table S4 in Supplementary Information).

Regarding the concentrations of the pollutants, we observe the largest declines in $PM_{10}$ in most of Europe, with an approximate annual decrease of 2.72%, followed by $NO_2$ (2.45%) and $PM_{2.5}$ (1.72%). Instead, we find that MDA8 $O_3$ rose by about 0.5% per year if the outlier representing year 2003 is excluded. These trends align with previous studies, which reported annual declines of around 1.7–2.2% in $NO_2$[23,24] and 1–2% in $PM_{2.5}$[14,24,25], as well as undefined trend between (−0.3 and +0.5%) in MDA8 $O_3$[24,26], over last two decades in Europe. These trends were also in agreement with[10] reanalysis products (CAMSRA and Merra-2) and ground-level observations[11], with an annual average decrease of 2.1–3.3% in $PM_{10}$, 2.3–2.5% in $NO_2$, 0.9–1.7% in $PM_{2.5}$ and a 0.1–0.9% annual increase in MDA8 $O_3$. Overall, our estimations provide further evidence of the slight upward trend of MDA8 $O_3$ in Europe over the last decades, when other pollutants decreased under the European Union's (EU) efforts to implement air quality control measures. Notably, this upward trend of MDA8 $O_3$ is latitudinally oriented, and largely related to temperatures and sunlight. These conditions promote the formation of $O_3$ from precursor pollutants like nitrogen oxides (NOx) and volatile organic compounds (VOCs). Previous studies[27–29] suggested that the reduction of NOx may have alleviated $O_3$ depletion in and around cities, particularly at night, due to lower titration of $O_3$ by NOx. Moreover, these studies underscore the necessity of prioritizing stronger control measures on VOCs over NOx for effective urban $O_3$ mitigation[27,28].

Previous analyses with WHO 2021 guideline[7–9,30] have primarily focused on urban areas, constrained by limited monitoring stations. These studies often faced inconsistencies in assessing different air pollutants against WHO guidelines, discrepancies in the availability of daily observations across different pollutants over space and time. Our study overcome these limits by providing full-coverage estimations covering European population exposure and time, enhancing a more thorough understanding of spatial and temporal disparities related to WHO guidelines. It highlighted a notable decrease in European population exposure to $PM_{2.5}$, $PM_{10}$ and $NO_2$, contrasting with the rise in MDA8 $O_3$ exposure. Additionally, the average exposure time and population exposed to unclean air areas for PM2.5 and O3 is much higher than for the other two pollutants, highlighting the urgency for

greater control for these pollutants. Furthermore, recent studies[31,32] have also linked MDA8 $O_3$ exposure exceeding WHO daily limits to substantial rises in hospital admissions for heart attacks, heart failure, and strokes. This further emphasizes the importance of addressing the increasing role of $O_3$ exposure, especially in the context of rapidly increasing threats from climate change in Europe.

Despite significant progress in reducing air pollution, our assessment found that over 85% of Europeans still experienced at least one day per year with compound event days, notably prevalent in Eastern and Western Europe (see Table S7 in Supplementary Information). It highlights the persistent need for heightened attention to exposure of compound event days. We also found that $PM_{2.5}$-$O_3$ days have become the second most prevalent category of compound event days in Europe, with their contribution increasing from 4.43% in 2004 to 35.23% in 2019. Recent increases in $PM_{2.5}$-$O_3$ days, especially in lower latitudes during warm seasons, are likely linked to climate change and complex interplay between $PM_{2.5}$ and $O_3$. Emission sources such as vehicle exhaust and industrial processes release both PM2.5 and $O_3$ precursors like VOCs and NOx. Global warming intensifies sunlight and raises temperature, particularly in summer, accelerating $O_3$ formation through photochemical reactions[33]. Subsequently, higher levels of $O_3$ will oxidize volatile organic gases or secondary organic aerosols in the atmosphere[34], leading to the condensation of certain oxidized compounds, ultimately forming secondary PM2.5 particles. Also, climate change increases the likelihood of wildfires, contributing to elevated $O_3$ and PM levels[35]. Lastly, biogenic VOCs (BVOCs) have been identified as second largest sources of the $O_3$ production in summer[36]. The emission rate of BVOCs also rises with increasing temperatures, reaching peak levels at around 38–40 °C[37], due to heightened metabolic activity in vegetation.

Ozone management presents a complex challenge due to its secondary formation pathway. Conventional air pollution control strategies, which focus on reducing primary pollutant emissions, may not be sufficient to effectively mitigate $O_3$ exceedances and associated compound event days. However, addressing climate change, which influences ozone formation through increased sunlight and rising temperatures, is crucial for long-term ozone management and protection of public health. This approach not only slows global warming but also curtails the rise of $O_3$ formation triggered by photochemical reactions in warmer seasons. Moreover, surface or tropospheric $O_3$, beyond impacting air quality, acts as a greenhouse gas. Its ability to absorb infrared radiation contributes to the trapping of heat in the lower atmosphere. By reducing tropospheric $O_3$ levels, we can help mitigate its role in the greenhouse effect, potentially breaking the cycle that leads to further $O_3$ generation. Implementing policies to prevent and manage wildfires can help in controlling the release of these compounds into the atmosphere, thereby reducing $O_3$ formation. Lastly, vehicles stand as the most prominent contributor to anthropogenic VOC emissions[36]. Implementing rigorous policies to control and diminish VOC emissions from vehicles can notably impact $O_3$ formation, particularly in urban areas characterized by dense vehicular traffic. Additionally, choosing low-BVOCs emission plants for urban green spaces also aids in mitigating BVOCs emissions, further improving air quality and reducing $O_3$ precursors.

while EEA[8] and WHO[7] provided varying compliance estimates for long-term unclean air populations in urban settings (see Table S9 in Supplementary Information), possibly due to differences in the analyzed urban areas, countries, periods or calculation methods, our analysis suggests around 98.10%, 80.15% and 86.34% of the population in the 35 European countries lived in 2019 in unsafe air areas for $PM_{2.5}$, $PM_{10}$ and $NO_2$, respectively. These results align closely with EEA's urban estimates of 97%, 81% and 94% for the 27 countries of the European Union (EU-27), respectively. However, our $NO_2$ estimates are more consisted with WHO's estimates in boarder human settlements settings, possibly due to wider inclusion of population beyond urban

areas in EEA's analysis. Notably, urban areas are more susceptible to experience higher $NO_2$, primarily driven by emissions from vehicles and residential sources[30,38]. Spatially, Northern Europe exhibits a significantly higher population proportion living in long-term clean air areas for $PM_{2.5}$, $PM_{10}$, and $NO_2$ compared to the rest of the continent (see Fig. S7 in the Supplementary Information). Additionally, with the introduction of the new long-term guideline for $O_3$ in 2021, the EEA[8,30] started to report non-compliance in all countries with the peak season $O_3$ standard in 2021 and 2022, which concurs with our findings from 2003 to 2019. These results underscore the significant improvements made in European air quality control for $PM_{10}$ and $NO_2$, while challenges in controlling $O_3$ levels underscore the need for a policy shift. Addressing global warming and air quality together with more comprehensive solutions is essential, requiring a macro perspective to collaborate with policymakers for effective action.

This study has several limitations worth acknowledging. Although we have conducted spatial and temporal cross-validations to assess the quality of our air pollution estimates, biases might persist due to the uneven distribution of ground-level stations and the limited number of observations in earlier periods. Also, the population exposure in this study does not include population changes within a year. Despite these limitations, our study serves as a solid foundation for future research and policy development addressing air quality management and public health concerns throughout Europe.

## Methods

### Exposure estimation and validation

In this study, we trained four separated Quantile LightGBM (QLG) models[21] for daily $PM_{2.5}$, $PM_{10}$, $NO_2$ and MDA8 $O_3$, with ground-level measurements from European environment information and observation network (Eionet). These four individual models for each pollutant are developed separately to maximize the information gathered from the varying numbers of background monitoring sites for each pollutant. Specifically, we used 1310, 2438, 1867 and 2021 sites for $PM_{2.5}$, $PM_{10}$, $NO_2$ and MDA8 $O_3$. To train these models, we gathered data from multiple sources, which are further described in the Supplementary Information (pp 3–6). The datasets encompassed various atmospheric aerosol data, like model predictions of size-resolved aerosol optical depth (AOD)[39], Atmospheric composition reanalysis data from CAMSRA[12] and MERRA-2[13]. Additionally, we incorporated reanalysis data from ERA5-land and ERA5[40], Gridded climate observations from E-OBS[41], Land use data, including road density data from GRIP global roads database[42], vegetation-related data from ERA5-land, Köppen-Geiger climate classification and local climate zone data from world urban dataset[43], Emission data from CAMSRA global emission inventories. These datasets spanned from January 1, 2003 to December 31, 2019, with different spatial and temporal resolutions provided in Table S1(Supplementary Information).

We computed daily averages if the data were originally available at hourly or 3-hourly resolution. To ensure consistent spatial resolutions, all continuous gridded data were bilinearly resampled to a horizontal resolution of 0.1° × 0.1°. For the Köppen–Geiger climate classification (approximately 0.08° × 0.08°), nearest neighbor interpolation was applied during resampling. Regarding the local climate zone data (1 km), resampling involved the use of the most frequent category. Subsequently, to align with the stations' observation, we extracted modeling data within a 0.05-degree buffer around each station's location. This extraction process employed the area-weighted average for continuous variables and the dominant category for categorical data.

To select the most relevant features for each air pollutant model, we employed the Boruta feature selection procedure[44]. This method considers interactions and nonlinear relationships during the selection of variables, making it robust and efficient for removing noise[45]. The selected variables for each air pollutant model are listed in

Table S2(Supplementary Information), and the 20 most important variables are listed in Figure S8(Supplementary Information).

As ground-level monitoring stations tend to be in and around urban areas[30,46], this may weaken the capacity of the model to estimate the concentrations in regions farther away from these sites, while causing overfitting in areas with higher station density. We employed a distance-weighted loss function (Supplementary Information pp9) during model training to address this issue and ensure that the model properly represents the areas with fewer monitoring sites. This approach involved assigning weights to the loss function based on the normalized distances between each site and its nearest neighboring sites. By doing so, we aimed to mitigate the potential biases associated with the non-uniform distribution of monitoring stations across the study area.

To evaluate the out-of-sample predictive capacity of the models, we used two different approaches. First, we randomly selected 10% of the ground-level sites as test sites to validate the model performance. Second, we conducted nested 5-fold cross-validation to obtain spatial and temporal out-of-sample predictions separately. For spatial out-of-sample predictions, we randomly divided the monitoring sites into five equal-sized subsamples. In each loop of predictions, four subsamples were used for model training and tuning, while the remaining subsample was used to obtain the out-of-sample predictions. For temporal out-of-sample predictions, we split the 17-year period into six subperiods consisting of three or two consecutive years. After obtaining these out-of-sample predictions, we calculated validation metrics (Supplementary Information pp9) such as the Pearson Correlation, NMB, and NRMSE to assess the performance of the models.

### Indicators calculation

We developed indicators describing three main aspects: (i) air pollution concentrations and the (ii) population count and (iii) cumulative time of exposure for individuals to air pollution values exceeding the guidelines (formulas and threshold of WHO guidelines (Table S3) are provided in the Supplementary Information, pp9–11):

We used the Re-Gridded Population of the World Version 4 (GPWv4) to calculate the daily PW regional concentrations for $PM_{2.5}$, $PM_{10}$, $NO_2$ and $O_3$ for all the grid-cells included in each of the 1426 NUTS3 regions. Further details are provided in the Supplementary Information (pp9). Changes over time were compared to annual PW concentrations with the Theil-Sen slope[10,24], which estimates the annual rate of change by taking the median of all possible pairwise slopes between data points. The Theil-Sen slope is less sensitive to outliers and is suitable for analyzing time series data with potential fluctuations and irregularities. Additionally, we used the Mann-Kendall test to evaluate the significance level of the trends[24].

We developed an indicator to represent the cumulative time of exposure for individuals to air pollution values exceeding the guidelines. Thus, the annual unclean air exposure time represents the PW average annual number of days in which the WHO daily limit for an air pollutant is exceeded. The indicator involves calculating person-days surpassing daily limits for each grid-cell annually, aggregated them over a region or set of regions (unit: person*day), and finally dividing by the total population of the region(s) (unit: day). The annual unclean air exposure time can be calculated for one individual air pollutant, or for two or more air pollutants simultaneously exceeding the WHO limits the same day and grid-cell (for compound event days). More details are provided in Supplementary Information (pp 10, 11).

To assess the population exposed to air pollution values in relation to established guidelines, we defined indicators called "Population in clean air areas" and "Population in unclean air areas", as the percentage of people living in areas where air quality either meets or exceeds recommended standards. Notably, 'clean air' aligned with WHO standards widespread adopted by many governments, balancing current expenditure and practicality against health benefits, but not imply entirely safe air level. For short-term guideline, we imposed that daily or 8h maximum values are met 99% of days in a given year, while for long-term limits, annual or peak season limits are not exceeded. Complementary criteria were used for unclean air areas. The mathematical formulation is described in the Supplementary Information (p10, 11).

## Data availability

The daily mean observations of PM2.5, PM10, NO2 and MDA8 O3 were collected from two main databases in European environment information and observation network (Eionet): the Airbase (2003–2012) and the Air Quality e-Reporting (2013–2019). The total AOD, Fine-mode AOD (fAOD), and Coarse-mode AOD (cAOD) products generated from our previous works[39]. Reanalysis meteorological data primarily came from the ERA5_land dataset. Air quality reanalysis data were collected from the CAMSRA. High-resolution gridded population data is from the Gridded Population of the World, Version 4 (GPWv4) database. The annual unclean air exposure time dataset generated in this study have been deposited in the github database [https://github.com/junesw2/Europepollu/tree/main/Sharedata] and are publicly available.

## Code availability

All analyses and visualizations in this study are facilitated by data and codes, which have been deposited in the Github [https://github.com/junesw2/Europepollu/tree/main]. https://doi.org/10.5281/zenodo.10551935. Other mapping and data processing are conducted using QGIS, R, and Python.

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

## Acknowledgements

Z.C., R.F.M.T., H.A. and J.B. gratefully acknowledge funding from the European Union's Horizon 2020 and Horizon Europe research and innovation programmes under grant agreement No 865564 (European Research Council Consolidator Grant EARLY-ADAPT, https://www.early-adapt.eu/), 101069213 (European Research Council Proof-of-Concept HHS-EWS) and 101123382 (European Research Council Proof-of-Concept FORECAST-AIR). Z.C. also acknowledges support from the grant PRE2020-091985 funded by "MCIN/AEI/10.13039/501100011033 and by European Social Fund invests in your future". H.A. also acknowledges funding from the European Union's Horizon Europe research and innovation programme under grant agreement No 101065876 (MSCA Postdoctoral Fellowship TEMP-MOMO). H.P. have received funding from the Ramon y Cajal grant (RYC2021-034511-I, MCIN/AEI/10.13039/501100011033 and European Union Next Generation EU/PRTR). H.P. and C.P.G.-P. acknowledge funding from the Ministerio para la Transición Ecológica y el Reto Demográfico (MITECO) as part of the Plan Nacional del Ozono project (BOE-A-2021-20183), Agencia Estatal de Investigacion (AEI) through the MITIGATE project (PID2020-116324RA695 I00/AEI/10.13039/501100011033), the AXA Research Fund, the European Research Council (ERC) under the Horizon 2020 research and innovation program through the ERC Consolidator Grant FRAGMENT (grant agreement no. 773051), and the Department of Research and Universities of the Government of Catalonia through the Atmospheric Composition Research Group (code 2021 SGR 01550).

ISGlobal authors acknowledge support from the grant CEX2018-000806-S funded by MCIN/AEI/10.13039/501100011033, and support from the Generalitat de Catalunya through the CERCA Program.

## Author contributions

In this project, Z.C., H.P., C.P.G-P., H.A. and J.B. conceptualized and acquired funding, while Z.C. and R.M. collected and processed the data. Z.C., H.P., C.P.G-P. and J.B. contributed to the methodology. Z.C. and J.B. prepared the initial draft of the paper. All authors revised the manuscript and approved the final version.

## Competing interests

The authors declare no competing interests.

## Additional information

**Supplementary information** The online version contains Supplementary Material available at https://doi.org/10.1038/s41467-024-46103-3.

