## [Peer Review File · Nature Communications]

Population exposure to multiple air pollutants and its compound episodes in EuropeReviewer #1 (Remarks to the Author):

The manuscript entitled "Population exposure to multiple air pollutants and its compound episodes in Europe" is timely, well-written, robust, and reports informative results (new insights) for scientific community, and policymakers. The QA/QC of the data is clearly presented.

Is the data (annual unsafe exposure time for each year & grid-cell) available online?

The state-of-the-art could be improved.

Minor revisions

Line 21: "...decreases of 2.72%, 2.45%, and 1.72%...", the scale should be mentioned (European scale?)

Line 21: "O3 levels increased" by?

Line 27: please report the version of the air quality guidelines (2005 or 2021)

Line 36: add a reference

Line 38: remove PM10. According to the latest EEA estimates, at least 238 000 people died prematurely in the EU in 2020 due to exposure to PM2.5 pollution above the WHO guideline level of 5 µg/m³.

Line 40: please cite "kind of effects"

Line 42: "... these pollutants... time scales," to be reformulated

Lines 49-50: unclear

Lines 64-66: add references

Line 69: add a ref such as De Marco et al., 2022 (10.1016/j.envres.2022.113048)

Lines 79-80: where are the ref #16-18?

Line 88: explain why you focus on PM2.5, NO2 and O3

Line 88: to be corrected as "data of daily xxx mean concentrations with meteorological..."

Line 105: cite the 4 models

Line 107: commonly we use MDA8 O3 (not O3), please correct throughout the manuscript

Line 155: a non-parametric test should be suited (e.g., Spearman rank test), did you check the data distribution?

Line 174: add a ref such as ref#27

Line 261-262: cross-check the units (for peak season: 60 µg m⁻³)

Line 337: surface ozone is rising in cities due to weakened ozone titration by NO (e.g., 10.1016/j.envres.2020.110193; 10.1016/j.envpol.2021.118690) but also PM2.5 decline

Line 381: usually when PM2.5 increase, surface ozone declines. How can you explain such PM2.5-O3 observation?

Line 385: "April to September" is the growing season at mid-latitude, with biogenic VOCs emissions leading to secondary air pollutants formation like fine particles

Line 391: as ozone is a secondary air pollutant, the control policies are not easy to implement to reduce ozone levels

Some typos should be corrected, e.g.,

lines 20 & 38, NO2 (subscript)

lines 385 & 394: PMP2.5-O3 (subscript)

Reviewer #2 (Remarks to the Author):

This is a well-written manuscript with new ways to curating results, and in general advances the field of knowledge. A few specific comments that the authors may consider include

line 60-67, it was suggested modelled data, either AOD or machine learning results, have all these advantages than ground measurements, without mentioning limitations/accuracies of models. The argument sounds slightly biased on ground measurements as a method/data source.

line 187-189, the wording of 'safe' and 'unsafe' can be misleading in the manuscript. As I understand, the WHO guideline value is a guideline value and it does not guarantee no health effects if the standard is met. Recent literature also supports health effects in areas of low air pollution levels. So the choice of word here need to be carefully considered.

line 294, authors claimed 'novel framework' although the three components individually were not new. It is fair to say the framework contributed to new knowledge yet different reviewers might have their own opinion on how novel it is.

Reviewer #3 (Remarks to the Author):

This study utilized machine learning models to estimate daily concentrations of PM_{2.5}, PM₁₀, NO₂, and O₃ at spatial resolution of 0.1° in 36 European countries. Based on the estimated results, the authors made a detailed analysis about the population exposure to air pollution considering short-term exposure, long term exposure and occurrence of compound episodes. The paper is well organized and the analysis is comprehensive. However, I perceive limitations in innovation, and the findings may not be sufficiently compelling to meet the criteria of NC. Many similar studies have been conducted in recent years with an even higher spatial resolution or broader scope; to cite a few, see references [1-3].

Here are some suggestions for the improvements of this manuscript.

Line 126-128, "all gridded data were bilinearly resampled". Categorical variables such as land-use, KÖPPEN-GEIGER climate classification, and local climate zone cannot be resampled using bilinear interpolation; instead, it requires the nearest neighbor interpolation method. Besides, you "trained the station data by aligning them with the nearest grid cell", have you considered there may be some stations located in the same grid cell? How do you handle this situation.

Line 138-142. Formulas should be provided. Have you tested the effect of the distance-weighted loss function through comparing it to the commonly used loss function? How much improvement can be achieved?

Line 155-156. Formulas should be provided.

Discussion. To long and not well-organized.

Figure 1. A scatter plot colored by point density would be better.

Other typos: Line 88, "Ozone levels" should be "ozone levels". Line 30, Abbreviation AOD not defined. Line 169, (p6) should be (pp6). Line 261-262, ug/cm³ should be µg/cm³. Line 282, Table S9. The citation of supplementary materials is inconsistent and not standardized. It is recommended to review the entire document and make corrections.

References:

- [1] Rentschler, J., & Leonova, N. (2023). Global air pollution exposure and poverty. *Nature Communications*, 14(1), 4432.
- [2] Stafoggia, M., Oftedal, B., Chen, J., Rodopoulou, S., Renzi, M., Atkinson, R. W., ... & Janssen, N. A. (2022). Long-term exposure to low ambient air pollution concentrations and mortality among 28 million people: results from seven large European cohorts within the ELAPSE project. *The Lancet Planetary Health*, 6(1), e9-e18.
- [3] Shaddick, G., Thomas, M. L., Mudu, P., Ruggeri, G., & Gummy, S. (2020). Half the world's population are exposed to increasing air pollution. *NPJ Climate and Atmospheric Science*, 3(1), 23.

REVIEWER COMMENTS

Reviewer #1 (Remarks to the Author):

The manuscript entitled "Population exposure to multiple air pollutants and its compound episodes in Europe" is timely, well-written, robust, and reports informative results (new insights) for scientific community, and policymakers. The QA/QC of the data is clearly presented.

Response: Thank you for your constructive and positive review of our manuscript.

Is the data (annual unsafe exposure time for each year & grid-cell) available online?

Response: Yes, we now provide the data in this link: <https://github.com/junesw2/Europepollu/tree/main/Sharedata>

The state-of-the-art could be improved.

Response: We appreciate your feedback on the state-of-the-art section and have taken your suggestions seriously. We have added the missing citation and some corrections, ensuring a more robust connection to the existing literature. Moreover, we have revisited the contents of the section to enhance clarity and comprehensiveness, particularly regarding the limitations of existing products or analysis, and the significance and necessity of our study.

Some examples like (section **introduction**, page 2-3, lines 43-77):

"To assess the threat posed by air pollution in Europe, recent compliance studies have predominantly relied on ground-based air pollutant monitoring networks⁷⁻⁹. However, these networks, concentrated primarily in urban areas, exhibit limited spatial coverage and fail to comprehensively represent the entire population. While ground-level measurements offer direct, accurate, and reliable real-world data, their spatial averaging and extrapolation introduces biases in exposure assessment. Additional limitations include frequently incomplete daily observation time series values, which can lead to biases when averaging observations from varying numbers of sites per day. Also, data availability from these networks is higher in more recent periods¹⁰, leading to inconsistencies in the prior analysis of multi-decadal concentrations changes.

Another key limitation pertains to the conventional analysis of guideline exceedances for each pollutant separately⁷⁻⁹. This approach overlooks occurrence of compound air pollution episodes, in which the WHO daily guidelines are simultaneously exceeded for two or more air pollutants. This is a noteworthy omission, as individuals may experience concentrations exceeding safe guidelines for multiple pollutants concurrently, potentially resulting in synergistic health effects that amplify overall health risks^{11,12}. Although some have begun exploring the interactive health impacts of co-exposure to specific combinations of pollutants, such as PM_{2.5} and O₃, further research on other combinations is imperative. Unfortunately, the unavailability of

consistent daily ground-level measurements for multiple air pollutants presents a challenge in comprehending the spatio-temporal patterns of population's co-exposure.

Using models constrained with observations represents a promising solution to these problems¹³. Global atmospheric composition reanalyses provide multidecadal daily estimates integrating a diverse range of satellite measurements^{14,15}. However, due to their coarse spatial resolution and the lack of integration of surface measurements, these datasets remain affected by significant biases at ground level. The use of air pollution models constrained by surface measurements over multidecadal periods, either for Europe or globally, have mostly focused on long-term averages (annual or monthly values)¹⁶⁻¹⁸, while models predicting daily concentrations have predominantly focused on a single pollutant, primarily PM_{2.5}^{19,20}. Consequently, consistent and accurate air pollution datasets allowing comprehensive understanding of population exposure to multiple air pollutants and its compound episodes in Europe is still lacking."

Minor revisions

Line 21: "...decreases of 2.72%, 2.45%, and 1.72%...", the scale should be mentioned (European scale?)

Response: Thanks, the text now reads (section **abstract**, page 1, lines 22-24):

"We observed a largest decline in European population-weighted PM₁₀ levels, followed by NO₂ and PM_{2.5}, with annual decreases of 2.72%, 2.45%, and 1.72%, respectively."

Line 21: "O₃ levels increased" by?

Response: Thanks, the text now reads (section **abstract**, page 1, lines 24):

"In contrast, O₃ levels increased by 0.58% in southern Europe, leading to a nearly fourfold rise in unclean air days."

Line 27: please report the version of the air quality guidelines (2005 or 2021)

Response: Thanks, the text now reads (section **abstract**, page 1, lines 18-22):

"Here, we estimated daily ambient PM_{2.5}, PM₁₀, NO₂, and O₃ concentrations at a 0.1-degree resolution across Europe during 2003-2019, to assess the occurrence of days exceeding WHO's 2021 guidelines (unclean air days) for one or multiple pollutants (compound days) in 1426 regions across 35 European countries, representing 543 million people."

Line 36: add a reference

Response: Done. (section **introduction**, page 2, lines 33):

" Air pollution poses a major health risk in Europe and worldwide^{1,2}. "

Ref:

1. European Environment Agency. *Europe's air quality status 2021*. (2022).
2. Institute for Health Metrics and Evaluation (IHME). *Global Burden of Disease Study 2019 (GBD 2019)*. (2020).

Line 38: remove PM10. According to the latest EEA estimates, at least 238 000 people died prematurely in the EU in 2020 due to exposure to PM2.5 pollution above the WHO guideline level of 5 µg/m³.

Response: Thanks for the correction, now we also updated to citation of the new report including 2021 data (Published 24 Nov 2023). Revised as (section **introduction**, page 2, lines 33-37):

“In 2021, the European Environment Agency (EEA) estimated over 253,000 premature deaths attributed to fine particulate matter (PM_{2.5}), 52,000 deaths to nitrogen dioxide (NO₂) and 22,000 deaths to ozone (O₃) exceeding the 2021 World Health Organization (WHO) annual limits³.”

Ref:

3. Harm to human health from air pollution in Europe: burden of disease 2023 — European Environment Agency. <https://www.eea.europa.eu/publications/harm-to-human-health-from-air-pollution/>.

Line 40: please cite “kind of effects”

Response: we now extended the example of chronic and acute exposure. Revised as (section **introduction**, page 2, lines 45-47):

“Exposure to air pollution, both chronic and acute, increases the risk of several cardiovascular and respiratory diseases, allergic reactions, diabetes, cognitive health disorders, and childhood development issues, among many others^{4,5}.”

Ref:

4. Mannucci, P. M., Harari, S., Martinelli, I. & Franchini, M. Effects on health of air pollution: a narrative review. *Intern Emerg Med* **10**, 657–662 (2015).
5. Kampa, M. & Castanas, E. Human health effects of air pollution. *Environmental pollution* **151**, 362–367 (2008).

Line 42: “... these pollutants... time scales,” to be reformulated

Response: Revised as (section **introduction**, page 2, lines 47-51):

“Recognizing these risks, in 2021, the WHO issued stricter air quality limits for each of these pollutants separately at different time scales, i.e. annual, peak season, 24 hours and daily maximum 8 hours, to mitigate both short-term and long-term health impacts caused by air pollutants.”

Lines 49-50: unclear

Response: Revised as (section **introduction**, page 2, lines 67-69):

“Individuals may experience concentrations exceeding safe guidelines for multiple pollutants concurrently, potentially resulting in synergistic health effects that amplify overall health risks^{9,10}.”

Lines 64-66: add references

Response: Added (section **introduction**, page 2, lines 60-62):

“Also, data availability from these networks is higher in recent periods¹¹, leading to

inconsistencies in the analysis of multi-decadal concentration changes. ”

Ref:

8. Lacima, A. *et al.* Long-term evaluation of surface air pollution in CAMSRA and MERRA-2 global reanalyses over Europe (2003-2020). *Geosci Model Dev* **16**, 2689–2718 (2023).

Line 69: add a ref such as De Marco et al., 2022 (10.1016/j.envres.2022.113048)

Response: Added, thanks. (section **introduction**, page 2, lines 67-68):

“Using models constrained with observations represents a promising solution to these problems⁹.”

Ref:

9. De Marco, A. *et al.* Ozone modelling and mapping for risk assessment: An overview of different approaches for human and ecosystems health. *Environ Res* **211**, 113048 (2022).

Lines 79-80: where are the ref #16-18?

Response: We revised the error of reference (section **introduction**, page 3, lines 79-82):

“Governments worldwide are increasingly acknowledging the necessity of addressing air pollutions collectively, such as the integrated control programs in the United States^{10,11}, due to their cost-benefit efficiency, as well as the significant benefits they offer in improving overall air quality and public health.”

Ref:

10. United States Environmental Protection Agency. Managing Air Quality - Multi-Pollutant Planning and Control | US EPA. 2023 <https://www.epa.gov/air-quality-management-process/managing-air-quality-multi-pollutant-planning-and-control#climate>.

11. Wesson, K., Fann, N., Morris, M., Fox, T. & Hubbell, B. A multi-pollutant, risk-based approach to air quality management: Case study for Detroit. *Atmos Pollut Res* **1**, 296–304 (2010).

Line 88: explain why you focus on PM2.5, NO2 and O3

Response: According to the EEA report, they rank among the top four contributors to mortality caused by air pollution in Europe. Revised as (section **introduction**, page 3, lines 90-93):

“This study uses Quantile LightGBM (QLG) machine learning models¹² to link ground-level station data of daily PM_{2.5}, PM₁₀, NO₂, and O₃ (the primary four air pollutants contributing to mortality³) mean concentrations with meteorological and air quality reanalysis data, aerosol optical depth (AOD) model estimations and ground-level emission data.”

Ref:

3. Harm to human health from air pollution in Europe: burden of disease 2023 — European Environment Agency. <https://www.eea.europa.eu/publications/harm-to-human-health-from-air-pollution/>.

Line 88: to be corrected as “data of daily xxx mean concentrations with meteorological…”

Response: Revised as (section **introduction**, page 3, lines 90-93):

“This study uses Quantile LightGBM (QLG) machine learning models¹² to link ground-level station data of daily PM_{2.5}, PM₁₀, NO₂, and O₃ (the primary four air pollutants contributing to mortality³) mean concentrations with meteorological and air quality reanalysis data, aerosol optical depth (AOD) model estimations and ground-level emission data.”

Line 105: cite the 4 models

Response: added (section **method**, page 9, lines 359-361):

“ In this study, we trained four separated Quantile LightGBM (QLG) models¹² with ground-level measurements of PM_{2.5}, PM₁₀, NO₂ and MDA8 O₃, from European environment information and observation network (Eionet).”

Ref:

12. Shi, Y., Ke, G., Chen, Z., Zheng, S. & Liu, T.-Y. Quantized Training of Gradient Boosting Decision Trees. in *Advances in Neural Information Processing Systems* (eds. Koyejo, S. et al.) vol. 35 18822–18833 (Curran Associates, Inc., 2022).

Line 107: commonly we use MDA8 O3 (not O3), please correct throughout the manuscript

Response: Done.

Line 155: a non-parametric test should be suiter (e.g., Spearman rank test), did you check the data distribution?

Response: Thanks for raising this issue. As PM_x and NO₂ data followed a right-tailed distribution, we also calculated the Spearman rank and Kendall tests (see Table 1). While these tests were considered, we ultimately chose the Pearson correlation due to its ability to detect linear relationships between observations and predictions, which is the desired relation we aim to obtain.

Table 1. Comparison in different correlation between observed and model-estimated PM_{2.5} (a), PM₁₀ (b), NO₂ (c), O₃ 8h max (d) concentrations from 2003 to 2019 in spatial cross-validation

	Pearson	Spearman	Kendall
PM _{2.5}	0.80	0.83	0.82
PM ₁₀	0.79	0.84	0.83
NO ₂	0.79	0.83	0.82
MDA8 O ₃	0.91	0.93	0.93

Line 174: add a ref such as ref#27

Response: Added, thanks. (section **method**, page 11, lines 430-431):

“Additionally, we used the Mann-Kendall test to evaluate the significance level of the trends¹³.”

Ref:

13. Sicard, P. *et al.* Trends in urban air pollution over the last two decades: A global perspective. *Science of The Total Environment* **858**, 160064 (2023).

Line 261-262: cross-check the units (for peak season: 60 $\mu\text{g m}^{-3}$)

Response: Thanks. Revised as (section **result**, page 5, lines 181-182):

“Regarding MDA8 O₃, almost no areas meet the WHO standard of 60 $\mu\text{g}/\text{m}^3$. Therefore, we adopted the WHO interim target 2 (70 $\mu\text{g}/\text{m}^3$) as the reference threshold, which showed no clear trend over the years (Figure 4d).”

Line 337: surface ozone is rising in cities due to weakened ozone titration by NO (e.g., 10.1016/j.envres.2020.110193; 10.1016/j.envpol.2021.118690) but also PM_{2.5} decline

Response: We agree that decreasing NO_x levels also affected ozone concentrations. We added this to our discussion, as (section **discussion**, page 7, lines 267-270):

“Previous studies¹⁴⁻¹⁶ suggested that the reduction of NO_x may have alleviated O₃ depletion in and around cities, particularly at night, due to lower titration of O₃ by NO_x. Moreover, these studies underscore the necessity of prioritizing stronger control measures on VOCs over NO_x for effective urban O₃ mitigation^{14,15}.”

Ref:

14. Sicard, P. *et al.* Ozone weekend effect in cities: Deep insights for urban air pollution control. *Environ Res* **191**, 110193 (2020).
15. Wang, N. *et al.* Aggravating O₃ pollution due to NO_x emission control in eastern China. *Science of the Total Environment* **677**, 732-744 (2019).
16. He, C. *et al.* The unexpected high frequency of nocturnal surface ozone enhancement events over China: characteristics and mechanisms. *Atmos Chem Phys* **22**, 15243-15261 (2022).

Line 381: usually when PM_{2.5} increase, surface ozone declines. How can you explain such PM_{2.5}-O₃ observation?

Response: Thank you for your interest in the relationship between PM_{2.5} and surface ozone levels. Their dynamic association is influenced by two main factors¹⁷: In winter, higher PM_{2.5} levels can weaken solar radiation, disrupting ozone formation. Conversely, during summer, elevated ozone levels indirectly contribute to secondary PM_{2.5} formation by oxidizing atmospheric compounds.¹⁸

We found these PM_{2.5}-O₃ events mainly occurring during the warm season, spanning from April to September (refer to Figure S4). In summer, despite the typically lower PM_{2.5} levels and stronger solar radiation compared to winter, the reduced likelihood of PM_{2.5} hindering solar radiation favors ozone formation during this season. Moreover, Emission sources such as vehicle exhaust and industrial processes release both PM_{2.5} and ozone precursors like VOCs and NO_x. Global warming intensifies sunlight and raises temperature in summer, accelerating O₃ formation through photochemical reactions¹⁹. Subsequently, higher levels of ozone will oxidize volatile organic gases or secondary organic aerosols in the atmosphere¹⁸, leading to the condensation of certain oxidized compounds, ultimately forming secondary PM_{2.5} particles. In general, the environmental

conditions linked to global warming, including intensified sunlight and higher summer temperatures, amplifying the challenge of elevated ozone and secondary PM formation over time¹⁹.

Also, climate change escalates the likelihood of wildfires, which heightens levels of both ozone and PM²⁰, further complicating their relationship during this season. These complexities in pollutant interactions and their sources challenge the conventional expectation of an inverse relationship between PM_{2.5} and surface ozone levels.

We also added this explanation in our discussion, as following (section **discussion**, page 8, lines 294-306):

“Recent increases in PM_{2.5}-O₃ days, especially in lower latitudes during warm seasons, are likely linked to climate change and complex interplay between PM_{2.5} and O₃. Emission sources such as vehicle exhaust and industrial processes release both PM_{2.5} and O₃ precursors like VOCs and NO_x. Global warming intensifies sunlight and raises temperature, particularly in summer, accelerating O₃ formation through photochemical reactions¹⁹. Subsequently, higher levels of O₃ will oxidize volatile organic gases or secondary organic aerosols in the atmosphere¹⁸, leading to the condensation of certain oxidized compounds, ultimately forming secondary PM_{2.5} particles. Also, climate change increases the likelihood of wildfires, contributing to elevated O₃ and PM levels²⁰. Lastly, biogenic VOCs (BVOCs) have been identified as second largest sources of the O₃ production in summer²¹. The emission rate of BVOCs also rises with increasing temperatures, reaching peak levels at around 38–40 °C²², due to heightened metabolic activity in vegetation.”

Ref:

19. Yin, Z., Wang, H., Li, Y., Ma, X. & Zhang, X. Links of climate variability in Arctic sea ice, Eurasian teleconnection pattern and summer surface ozone pollution in North China. *Atmos Chem Phys* **19**, 3857–3871 (2019).
20. Jaffe, D. A. & Wigder, N. L. Ozone production from wildfires: A critical review. *Atmos Environ* **51**, 1–10 (2012).
21. Zhan, J. *et al.* The contribution of industrial emissions to ozone pollution: identified using ozone formation path tracing approach. *NPJ Clim Atmos Sci* **6**, 37 (2023).
22. Janyasuthiwong, S. *et al.* Biogenic volatile organic compound emission from tropical plants in relation to temperature changes. *Environmental Challenges* **9**, 100643 (2022).

Line 385: “April to September” is the growing season at mid-latitude, with biogenic VOCs emissions leading to secondary air pollutants formation like fine particles

Response: Thanks for raising this new angle. Indeed, it can be one of potential reasons why PM_{2.5}-O₃ events become more frequent. Janyasuthiwong *et al.* (2022) found out biogenic VOCs (BVOCs) emission rate increased with increasing temperature up to 38–40 °C and decreased after²². Elevated temperature, sunlight and growing season for plants in warm season can increase metabolic activity in vegetation, resulting in higher BVOCs emissions. VOCs from both natural and anthropogenic sources play important roles in the formation of ozone (O₃) and secondary components of fine particulate matter (PM_{2.5})²³, especially

when they react with other pollutants in the presence of sunlight. Zhan et al. (2023) also proved their roles and found VOCs from vehicle exhaust and BVOCs are two largest sources of the O₃ production in summer²¹.

Line 391: as ozone is a secondary air pollutant, the control policies are not easy to implement to reduce ozone levels

Response: We agree that addressing ozone levels poses challenges, yet viable solutions exist. Firstly, focusing on controlling greenhouse gases emission pivotal to climate change is essential. This approach not only slows global warming, but also curtails the rise of ozone formation triggered by photochemical reactions in warmer seasons. Moreover, surface or tropospheric ozone, beyond impacting air quality, acts as a greenhouse gas. Its ability to absorb infrared radiation contributes to the trapping of heat in the lower atmosphere. By reducing tropospheric ozone levels, we can help mitigate its role in the greenhouse effect, potentially breaking the cycle that leads to further ozone generation.

Wildfires release significant amounts of pollutants, including volatile organic compounds (VOCs), which are precursors to ozone formation. Implementing policies to prevent and manage wildfires can help in controlling the release of these compounds into the atmosphere, thereby reducing ozone formation.

Vehicles stand as prominent contributors to VOC emissions²¹. Implementing rigorous policies to control and diminish VOC emissions from vehicles can notably impact ozone formation, particularly in urban areas characterized by dense vehicle traffic. Studies underscore the need for ozone mitigation, especially in urban settings where most people live, to prioritize stronger controls on VOCs over NO_x, emphasizing the importance of this targeted approach.

We added these solutions in our discussion, as follows (section **discussion**, page 8, lines 308-326):

“Ozone management presents a complex challenge due to its secondary formation pathway. Conventional air pollution control strategies, which focus on reducing primary pollutant emissions, may not be sufficient to effectively mitigate O₃ exceedances and associated compound days. However, addressing climate change, which influences ozone formation through increased sunlight and rising temperatures, is crucial for long-term ozone management and protection of public health. This approach not only slows global warming but also curtails the rise of O₃ formation triggered by photochemical reactions in warmer seasons. Moreover, surface or tropospheric O₃, beyond impacting air quality, acts as a greenhouse gas. Its ability to absorb infrared radiation contributes to the trapping of heat in the lower atmosphere. By reducing tropospheric O₃ levels, we can help mitigate its role in the greenhouse effect, potentially breaking the cycle that leads to further O₃ generation. Implementing policies to prevent and manage wildfires can help in controlling the release of these compounds into the atmosphere, thereby reducing O₃ formation. Lastly, vehicles stand as the most prominent contributor to VOC emissions²¹. Implementing rigorous policies to control and diminish VOC emissions from vehicles can notably impact O₃ formation, particularly in urban areas characterized by dense vehicular traffic.

Additionally, choosing low-BVOCs emission plants for urban green spaces also aids in mitigating BVOCs emissions, further improving air quality and reducing O₃ precursors.”

Some typos should be corrected, e.g.,
lines 20 & 38, NO₂ (subscript)

Done

lines 385 & 394: PMP2.5-O₃ (subscript)

Done

References:

1. European Environment Agency. *Europe's air quality status 2021*. (2022).
2. Institute for Health Metrics and Evaluation (IHME). *Global Burden of Disease Study 2019 (GBD 2019)*. (2020).
3. Harm to human health from air pollution in Europe: burden of disease 2023 — European Environment Agency. <https://www.eea.europa.eu/publications/harm-to-human-health-from-air-pollution/>.
4. Mannucci, P. M., Harari, S., Martinelli, I. & Franchini, M. Effects on health of air pollution: a narrative review. *Intern Emerg Med* **10**, 657–662 (2015).
5. Kampa, M. & Castanas, E. Human health effects of air pollution. *Environmental pollution* **151**, 362–367 (2008).
6. Dominici, F., Peng, R. D., Barr, C. D. & Bell, M. L. Protecting human health from air pollution: shifting from a single-pollutant to a multi-pollutant approach. *Epidemiology* **21**, 187 (2010).
7. Mauderly, J. L. & Samet, J. M. Is there evidence for synergy among air pollutants in causing health effects? *Environ Health Perspect* **117**, 1–6 (2009).
8. Lacima, A. *et al.* Long-term evaluation of surface air pollution in CAMSRA and MERRA-2 global reanalyses over Europe (2003-2020). *Geosci Model Dev* **16**, 2689–2718 (2023).
9. De Marco, A. *et al.* Ozone modelling and mapping for risk assessment: An overview of different approaches for human and ecosystems health. *Environ Res* **211**, 113048 (2022).
10. United States Environmental Protection Agency. Managing Air Quality - Multi-Pollutant Planning and Control | US EPA. 2023 <https://www.epa.gov/air-quality-management-process/managing-air-quality-multi-pollutant-planning-and-control#climate>.
11. Wesson, K., Fann, N., Morris, M., Fox, T. & Hubbell, B. A multi-pollutant, risk-based approach to air quality management: Case study for Detroit. *Atmos Pollut Res* **1**, 296–304 (2010).
12. Shi, Y., Ke, G., Chen, Z., Zheng, S. & Liu, T.-Y. Quantized Training of Gradient Boosting Decision Trees. in *Advances in Neural Information Processing Systems* (eds. Koyejo, S. *et al.*) vol. 35 18822–18833 (Curran Associates, Inc., 2022).
13. Sicard, P. *et al.* Trends in urban air pollution over the last two decades: A global

- perspective. *Science of The Total Environment* **858**, 160064 (2023).
14. Sicard, P. *et al.* Ozone weekend effect in cities: Deep insights for urban air pollution control. *Environ Res* **191**, 110193 (2020).
 15. Wang, N. *et al.* Aggravating O₃ pollution due to NO_x emission control in eastern China. *Science of the Total Environment* **677**, 732–744 (2019).
 16. He, C. *et al.* The unexpected high frequency of nocturnal surface ozone enhancement events over China: characteristics and mechanisms. *Atmos Chem Phys* **22**, 15243–15261 (2022).
 17. Jia, M. *et al.* Inverse relations of PM_{2.5} and O₃ in air compound pollution between cold and hot seasons over an urban area of east China. *Atmosphere (Basel)* **8**, 59 (2017).
 18. Hodan, W. M. & Barnard, W. R. Evaluating the contribution of PM_{2.5} precursor gases and re-entrained road emissions to mobile source PM_{2.5} particulate matter emissions. *MACTEC Federal Programs, Research Triangle Park, NC* (2004).
 19. Yin, Z., Wang, H., Li, Y., Ma, X. & Zhang, X. Links of climate variability in Arctic sea ice, Eurasian teleconnection pattern and summer surface ozone pollution in North China. *Atmos Chem Phys* **19**, 3857–3871 (2019).
 20. Jaffe, D. A. & Wigder, N. L. Ozone production from wildfires: A critical review. *Atmos Environ* **51**, 1–10 (2012).
 21. Zhan, J. *et al.* The contribution of industrial emissions to ozone pollution: identified using ozone formation path tracing approach. *NPJ Clim Atmos Sci* **6**, 37 (2023).
 22. Janyasuthiwong, S. *et al.* Biogenic volatile organic compound emission from tropical plants in relation to temperature changes. *Environmental Challenges* **9**, 100643 (2022).
 23. Ren, Y. *et al.* Air quality and health effects of biogenic volatile organic compounds emissions from urban green spaces and the mitigation strategies. *Environmental Pollution* **230**, 849–861 (2017).

Reviewer #2 (Remarks to the Author):

This is a well-written manuscript with new ways to curating results, and in general advances the field of knowledge. A few specific comments that the authors may consider include

Response: Thank you for taking the time to review our manuscript and for your constrictive and positive feedback.

line 60-67, it was suggested modelled data, either AOD or machine learning results, have all these advantages than ground measurements, without mentioning limitations/accuracies of models. The argument sounds slightly biased on ground measurements as a method/data source.

Response: We added and revised some contents to highlight the existing issues with models (section **introduction**, page 2-3, lines 67-77):

“Using models constrained with observations represents a promising solution to these problems¹³. Global atmospheric composition reanalyses provide multidecadal daily estimates integrating a diverse range of satellite measurements^{14,15}. However, due to their coarse spatial resolution and the lack of integration of surface measurements, these datasets remain affected by significant biases at ground level. The use of air pollution models constrained by surface measurements over multidecadal periods, either for Europe or globally, have mostly focused on long-term averages (annual or monthly values)¹⁶⁻¹⁸, while models predicting daily concentrations have predominantly focused on a single pollutant, primarily PM_{2.5}^{19,20}. Consequently, consistent and accurate air pollution datasets allowing comprehensive understanding of population exposure to multiple air pollutants and its compound episodes in Europe is still lacking.”

line 187-189, the wording of 'safe' and 'unsafe' can be misleading in the manuscript. As I understand, the WHO guideline value is a guideline value and it does not guarantee no health effects if the standard is met. Recent literature also supports health effects in areas of low air pollution levels. So the choice of word here need to be carefully considered.

Response: Thanks for comment and suggestions, we agree that terms like "safe" and "unsafe" might be oversimplifying and misleading. We replaced them by “clean-air” and “unclean-air”. This change acknowledges the widespread adoption of WHO standards by many governments as their primary control policy benchmark.

line 294, authors claimed 'novel framework' although the three components individually were not new. It is fair to say the framework contributed to new knowledge yet different reviewers might have their own opinion on how novel it is.

Response: We appreciate and acknowledge this useful feedback regarding the words 'novel framework.' Thus, we have deleted “novel framework”, and updated our description to emphasize the framework's contribution to advancing knowledge. We also

further underscored the main novelty of our study in two key aspects: Firstly, our findings offer comprehensive evidences of both short-term and long-term exposure to main air pollutants continent-wide, surpassing urban boundaries. Additionally, it introduces valuable insights into compound episodes involving these pollutants, contributing significantly to our understanding and management of air pollution.

Revised as follows (section **Discussion**, page 6, lines 217-233):

“In general, our study provides a comprehensive assessment of spatial and temporal inequities in population exposure to air pollutants in 1426 regions across 35 European countries, representing 543 million people. Our findings reveal a substantial reduction in European population exposure to most air pollutants. However, PM_{2.5} and O₃ levels continue to surpass WHO guidelines in numerous regions, resulting in a relatively higher number of people exposed to unclean air levels. Moreover, our pioneer assessment of compound event days showed annual occurrence of compound event days decreased from 78.5 to 17.4 days over 2003-2019, but over 86.3% of the European population still experienced at least one compound event days per year in 2012-2019. PM_{2.5}-O₃ was the only compound event days that increased and became the second most frequent type of compounds in Europe during 2012-2019. Overall, our findings present comprehensive evidences of both short and long-term exposure to the main pollutants with largest impact on human health and mortality, by performing an exhaustive continental-wide regional analysis not restricted to urban settings only. Additionally, it introduces valuable insights into compound event days involving these pollutants, significantly enriching our understanding of multi-hazard exposure, and potentially guiding air pollution management policies.”

References:

8. Lacima, A. *et al.* Long-term evaluation of surface air pollution in CAMSRA and MERRA-2 global reanalyses over Europe (2003-2020). *Geosci Model Dev* **16**, 2689–2718 (2023).
25. Exceedance of air quality standards in Europe. Exceedance of air quality standards in Europe. 2023 <https://www.eea.europa.eu/ims/exceedance-of-air-quality-standards>.
26. Bowdalo, D. *et al.* Compliance with 2021 WHO air quality guidelines across Europe will require radical measures. *Environmental Research Letters (ERL)* **17**, (2022).
27. Hammer, M. S. *et al.* Global estimates and long-term trends of fine particulate matter concentrations (1998–2018). *Environ Sci Technol* **54**, 7879–7890 (2020).
28. Shaddick, G. *et al.* Data integration model for air quality: a hierarchical approach to the global estimation of exposures to ambient air pollution. *J R Stat Soc Ser C Appl Stat* **67**, 231–253 (2018).

Reviewer #3 (Remarks to the Author):

This study utilized machine learning models to estimate daily concentrations of PM_{2.5}, PM₁₀, NO₂, and O₃ at spatial resolution of 0.1° in 36 European countries. Based on the estimated results, the authors made a detailed analysis about the population exposure to air pollution considering short-term exposure, long term exposure and occurrence of compound episodes. The paper is well organized and the analysis is comprehensive.

Response: Thank you for taking the time to review our manuscript and for your constrictive and positive feedback.

However, I perceive limitations in innovation, and the findings may not be sufficiently compelling to meet the criteria of NC. Many similar studies have been conducted in recent years with an even higher spatial resolution or broader scope; to cite a few, see references [1-3].

Response: We appreciate your consideration regarding the novelty of our study. Our research introduces two significant novel aspects that distinguish it from existing studies. Firstly, it offers a comprehensive evaluation and new understandings in both daily (short-term) and long-term exposure to four key air pollutants across the entire European continent, extending beyond the urban setting. Secondly, it contributes to new insights on compound episodes involving these four key air pollutants.

Here we compare our study with the 3 examples provided by the reviewer:

[1] Rentschler, J., & Leonova, N. (2023). Global air pollution exposure and poverty. *Nature Communications*, 14(1), 4432.

It is a global scale study but only focuses on PM_{2.5}. The time scale of their PM_{2.5} estimates only considers annual and monthly values. This is explained in the article (section **Introduction**, page 2, lines 71-77):

“The use of air pollution models constrained by surface measurements over multidecadal periods, either for Europe or globally, have mostly focused on long-term averages (annual or monthly values)^{16–18}, while models predicting daily concentrations have predominantly focused on a single pollutant, primarily PM_{2.5}^{19,20}. Consequently, consistent and accurate air pollution datasets allowing comprehensive understanding of population exposure to multiple air pollutants and its compound episodes in Europe is still lacking.”

[2] Stafoggia, M., Oftedal, B., Chen, J., Rodopoulou, S., Renzi, M., Atkinson, R. W., ... & Janssen, N. A. (2022). Long-term exposure to low ambient air pollution concentrations and mortality among 28 million people: results from seven large European cohorts within the ELAPSE project. *The Lancet Planetary Health*, 6(1), e9-e18.

It is a multipollutant cohort study based on six countries and one city (Belgium, Denmark, England, the Netherlands, Norway, Switzerland and Rome (Italy)). These six areas are high-income countries mostly in northern or western Europe, and so they might not fully represent the entire European population. Furthermore, their exposure assessment is only available at the annual scale, and it does not analyze the new WHO guidelines.

[3] Shaddick, G., Thomas, M. L., Mudu, P., Ruggeri, G., & Gumy, S. (2020). Half the world's population are exposed to increasing air pollution. *NPJ Climate and Atmospheric Science*, 3(1), 23.

It is global study but it is only based on annual estimates of PM_{2.5}.

Overall, to our best knowledge, our study fills a gap in providing comprehensive evaluation in daily and long-term exposure of four key air pollutants across the entire European continent. Furthermore, most studies neglect occurrence of compound days, in which the WHO daily guidelines are simultaneously exceeded for two or more air pollutants. However, Compound days pose a significant health concern, as individuals are exposed to a combination of pollutants that can interact in complex ways, leading to the formation of secondary pollutants⁶ or synergistic health effects⁷. Before conducting more precise health assessment associated with specific combinations of pollutants, we need to have enough insights from compound days analysis (the frequency and trends of compound days) rather than only with single pollutant. These insights also enable us to develop targeted interventions on reducing specific pollutant combinations to effectively improve public health.

Moreover, the revised manuscript, especially the discussion part, highlights the primary challenge and complexities in air quality control under climate change, emphasizing the need for a multifaceted approach that tackles both global warming and air pollution. Especially while improvements have been made for PM₁₀ and NO₂, some secondary pollutants like O₃ presents a persistent challenge, requiring a policy shift and collaboration with policymakers to implement comprehensive solutions. We are confident that these comprehensive findings and distinctive insights contribute significantly to the broader scientific landscape.

Here are some suggestions for the improvements of this manuscript.

Line 126-128, "all gridded data were bilinearly resampled". Categorical variables such as land-use, KÖPPEN-GEIGER climate classification, and local climate zone cannot be resampled using bilinear interpolation; instead, it requires the nearest neighbor interpolation method.

Response: Thank you for highlighting the discrepancy in our description. Sorry for the unclear description, we have updated the description to accurately reflect the resampling techniques used for different data types, as follows (section Method, page 10, lines 379-383):

"To ensure consistent spatial resolutions, all continuous gridded data were bilinearly resampled to a horizontal resolution of 0.1° x 0.1°. For the Köppen-Geiger climate classification (approximately 0.08° x 0.08°), nearest neighbor interpolation was applied during resampling. Regarding the local climate zone data (1km), resampling involved the use of the most frequent category."

Besides, you "trained the station data by aligning them with the nearest grid cell", have you

considered there may be some stations located in the same grid cell? How do you handle this situation.

Response: We revised the script to give more details on how we aligned the observation with grid cells. The text now reads (section Method, page 10, lines 383-386):

" Subsequently, to align with the stations' observation, we extracted modeling data within a 0.05-degree buffer around each station's location. This extraction process employed the area-weighted average for continuous variables and the dominant category for categorical data."

Furthermore, we also applied a distance-weighted loss function to mitigate the potential overfit in areas with higher station density.

Line 138-142. Formulas should be provided. Have you tested the effect of the distance-weighted loss function through comparing it to the commonly used loss function? How much improvement can be achieved?

Response: We added the formula in the appendix (page 9, Line 111-117) "The mathematical formulation for distance-weighted loss function represented as:

$$W_i = \frac{(D_i - D_{min})}{(\bar{D} - D_{min})}$$

$$L(Y, Y^*) = \sum_{i=1}^n W_i * L(y_i, y_i^*)$$

where D_i is the distance of station i to its nearest site; D_{min} and \bar{D} denote the minimum and average of D_i in Europe; $L(Y, Y^*)$ is the overall loss function and $L(y_i, y_i^*)$ the loss functions for station i (y_i and y_i^* are the observations and predictions at station i). "

Moreover, we have conducted a comparison between the models using the common loss function and the one with the distance-weighted loss function, and the results are provided in Table 2. The models employing the distance-weighted loss function outperformed those with common loss function in spatial cross-validation. This comparison highlights the effectiveness of the distance-weighted approach in enhancing our model's performance.

Table 2. Comparison in spatial cross-validation between origin model and model with distance weighted loss function

		PM2.5	PM10	NO2	O3
original	Corr	0.77	0.77	0.76	0.88
	NMB(%)	-0.8	-3.76	-2.34	0.05
	NRMSE(%)	2.06	3.24	9.64	4.52
distance weighted loss function	Corr	0.80	0.79	0.79	0.90
	NMB(%)	-0.6	-3.81	-1.99	0.02
	NRMSE(%)	1.84	2.07	8.99	3.35

Line 155-156. Formulas should be provided.

Response: We added the formula in appendices (page 9, Line 119-123)

“**Validation metrics:**

$$\text{Normalized Mean Bias (NMB \%)} = \frac{\sum_{k=1}^n (\text{Pred}_k - \text{Obs}_k)}{\sum_{k=1}^n (\text{Obs}_k)}$$

$$\text{Normalized Root Mean Square Error (NRMSE \%)} = \frac{\sqrt{\frac{\sum_{k=1}^n (\text{Pred}_k - \text{Obs}_k)^2}{N}}}{IQR(\text{Obs})}$$

Where Pred is the model prediction; Obs is the observation and n is the total numbers of observations.”

Discussion. To long and not well-organized.

Response: Thanks for your concerns on our discussion. We revised them to make them more compact and logical.

Moreover, the revised manuscript put more emphasis on the explanation of our findings like (section **Discussion**, page 7, lines 264-270):

“Notably, this upward trend of MDA8 O₃ is latitudinally oriented, and largely related to temperatures and sunlight. These conditions promote the formation of O₃ from precursor pollutants like nitrogen oxides (NO_x) and volatile organic compounds (VOCs). Previous studies³⁰⁻³² suggested that the reduction of NO_x may have alleviated O₃ depletion in and around cities, particularly at night, due to lower titration of O₃ by NO_x. Moreover, these studies underscore the necessity of prioritizing stronger control measures on VOCs over NO_x for effective urban O₃ mitigation^{30,31}.”

We also emphasize the primary challenge in air quality control amid climate change and proposes potential measures to alleviate the escalation in ozone levels, like (section **Discussion**, page 8, lines 294-306):

“Recent increases in PM_{2.5}-O₃ days, especially in lower latitudes during warm seasons, are likely linked to climate change and complex interplay between PM_{2.5} and O₃. Emission sources such as vehicle exhaust and industrial processes release both PM_{2.5} and O₃ precursors like VOCs and NO_x. Global warming intensifies sunlight and raises temperature, particularly in summer, accelerating O₃ formation through photochemical reactions³⁶. Subsequently, higher levels of O₃ will oxidize volatile organic gases or secondary organic aerosols in the atmosphere³⁷, leading to the condensation of certain oxidized compounds, ultimately forming secondary PM_{2.5} particles. Also, climate change

increases the likelihood of wildfires, contributing to elevated O₃ and PM levels³⁸. Lastly, biogenic VOCs (BVOCs) have been identified as second largest sources of the O₃ production in summer³⁹. The emission rate of BVOCs also rises with increasing temperatures, reaching peak levels at around 38–40 °C⁴⁰, due to heightened metabolic activity in vegetation.“

(section **Discussion**, page 8, lines 308-326):

“Ozone management presents a complex challenge due to its secondary formation pathway. Conventional air pollution control strategies, which focus on reducing primary pollutant emissions, may not be sufficient to effectively mitigate O₃ exceedances and associated compound event days. However, addressing climate change, which influences ozone formation through increased sunlight and rising temperatures, is crucial for long-term ozone management and protection of public health. This approach not only slows global warming but also curtails the rise of O₃ formation triggered by photochemical reactions in warmer seasons. Moreover, surface or tropospheric O₃, beyond impacting air quality, acts as a greenhouse gas. Its ability to absorb infrared radiation contributes to the trapping of heat in the lower atmosphere. By reducing tropospheric O₃ levels, we can help mitigate its role in the greenhouse effect, potentially breaking the cycle that leads to further O₃ generation. Implementing policies to prevent and manage wildfires can help in controlling the release of these compounds into the atmosphere, thereby reducing O₃ formation. Lastly, vehicles stand as the most prominent contributor to anthropogenic VOC emissions³⁹. Implementing rigorous policies to control and diminish VOC emissions from vehicles can notably impact O₃ formation, particularly in urban areas characterized by dense vehicular traffic. Additionally, choosing low-BVOCs emission plants for urban green spaces also aids in mitigating BVOCs emissions, further improving air quality and reducing O₃ precursors.”

Figure 1. A scatter plot colored by point density would be better.

Response: Revised, thanks.

Figure 1. Comparison between observed and model-estimated PM_{2.5} (a), PM₁₀ (b), NO₂ (c), O₃ 8h max (d) concentrations from 2003 to 2019 in spatial cross-validation

Other typos:

Line 88, "Ozone levels" should be "ozone levels".

Response: Revised, thanks.

Line 30, Abbreviation AOD not defined.

Response: Done, revised as (section **introduction**, page 3, lines 90-93):

"This study uses Quantile LightGBM (QLG) machine learning models¹² to link ground-level station data of daily PM_{2.5}, PM₁₀, NO₂, and O₃ (the primary four air pollutants contributing to mortality³) mean concentrations with meteorological and air quality reanalysis data, aerosol optical depth (AOD) model estimations and ground-level emission data."

Line 169, (p6) should be (pp6).

Response: Revised.

Line 261-262, ug/cm³ should be ug/m³.

Response: Revised.

Line 282, The citation of supplementary materials is inconsistent and not standardized. It is recommended to review the entire document and make corrections.

Response: Thank you for bringing this to our attention. We have taken the necessary steps to revise the format of the supplementary materials as per your suggestion.

References:

1. European Environment Agency. *Europe's air quality status 2021*. (2022).
2. Institute for Health Metrics and Evaluation (IHME). *Global Burden of Disease Study 2019 (GBD 2019)*. (2020).
3. Harm to human health from air pollution in Europe: burden of disease 2023 — European Environment Agency. <https://www.eea.europa.eu/publications/harm-to-human-health-from-air-pollution/>.
4. Mannucci, P. M., Harari, S., Martinelli, I. & Franchini, M. Effects on health of air pollution: a narrative review. *Intern Emerg Med* **10**, 657–662 (2015).
5. Kampa, M. & Castanas, E. Human health effects of air pollution. *Environmental pollution* **151**, 362–367 (2008).
6. Dominici, F., Peng, R. D., Barr, C. D. & Bell, M. L. Protecting human health from air pollution: shifting from a single-pollutant to a multi-pollutant approach. *Epidemiology* **21**, 187 (2010).
7. Mauderly, J. L. & Samet, J. M. Is there evidence for synergy among air pollutants in causing health effects? *Environ Health Perspect* **117**, 1–6 (2009).
8. Lacima, A. *et al.* Long-term evaluation of surface air pollution in CAMSRA and MERRA-2 global reanalyses over Europe (2003-2020). *Geosci Model Dev* **16**, 2689–2718 (2023).
9. United States Environmental Protection Agency. Managing Air Quality - Multi-Pollutant Planning and Control | US EPA. 2023 <https://www.epa.gov/air-quality-management-process/managing-air-quality-multi-pollutant-planning-and-control#climate>.
10. Wesson, K., Fann, N., Morris, M., Fox, T. & Hubbell, B. A multi-pollutant, risk-based approach to air quality management: Case study for Detroit. *Atmos Pollut Res* **1**, 296–304 (2010).
11. Shi, Y., Ke, G., Chen, Z., Zheng, S. & Liu, T.-Y. Quantized Training of Gradient Boosting Decision Trees. in *Advances in Neural Information Processing Systems* (eds. Koyejo, S. *et al.*) vol. 35 18822–18833 (Curran Associates, Inc., 2022).
12. Sicard, P. *et al.* Ozone weekend effect in cities: Deep insights for urban air pollution control. *Environ Res* **191**, 110193 (2020).
13. Wang, N. *et al.* Aggravating O₃ pollution due to NO_x emission control in eastern China. *Science of the Total Environment* **677**, 732–744 (2019).

14. He, C. *et al.* The unexpected high frequency of nocturnal surface ozone enhancement events over China: characteristics and mechanisms. *Atmos Chem Phys* **22**, 15243–15261 (2022).
15. Jia, M. *et al.* Inverse relations of PM_{2.5} and O₃ in air compound pollution between cold and hot seasons over an urban area of east China. *Atmosphere (Basel)* **8**, 59 (2017).
16. Hodan, W. M. & Barnard, W. R. Evaluating the contribution of PM_{2.5} precursor gases and re-entrained road emissions to mobile source PM_{2.5} particulate matter emissions. *MACTEC Federal Programs, Research Triangle Park, NC* (2004).
17. Yin, Z., Wang, H., Li, Y., Ma, X. & Zhang, X. Links of climate variability in Arctic sea ice, Eurasian teleconnection pattern and summer surface ozone pollution in North China. *Atmos Chem Phys* **19**, 3857–3871 (2019).
18. Jaffe, D. A. & Wigder, N. L. Ozone production from wildfires: A critical review. *Atmos Environ* **51**, 1–10 (2012).
19. Zhan, J. *et al.* The contribution of industrial emissions to ozone pollution: identified using ozone formation path tracing approach. *NPJ Clim Atmos Sci* **6**, 37 (2023).
20. Janyasuthiwong, S. *et al.* Biogenic volatile organic compound emission from tropical plants in relation to temperature changes. *Environmental Challenges* **9**, 100643 (2022).
21. Ren, Y. *et al.* Air quality and health effects of biogenic volatile organic compounds emissions from urban green spaces and the mitigation strategies. *Environmental Pollution* **230**, 849–861 (2017).
22. Hammer, M. S. *et al.* Global estimates and long-term trends of fine particulate matter concentrations (1998–2018). *Environ Sci Technol* **54**, 7879–7890 (2020).
23. Shaddick, G. *et al.* Data integration model for air quality: a hierarchical approach to the global estimation of exposures to ambient air pollution. *J R Stat Soc Ser C Appl Stat* **67**, 231–253 (2018).
24. Southerland, V. A. *et al.* Global urban temporal trends in fine particulate matter (PM_{2.5}) and attributable health burdens: estimates from global datasets. *Lancet Planet Health* **6**, e139–e146 (2022).
25. Lary, D. J. *et al.* Estimating the global abundance of ground level presence of particulate matter (PM_{2.5}). *Geospat Health* **8**, S611 (2014).
26. Yu, W. *et al.* Global estimates of daily ambient fine particulate matter concentrations and unequal spatiotemporal distribution of population exposure: a machine learning modelling study. *Lancet Planet Health* **7**, e209–e218 (2023).